# Consistency-Based Semi-Supervised Active Learning: Towards Minimizing Labeling Budget

## Abstract

Active learning (AL) integrates data labeling and model training to minimize the labeling cost by prioritizing the selection of high value data that can best improve model performance. Readily-available unlabeled data are used for selection mechanisms, but are not used for model training in most conventional pool-based AL methods. To minimize the labeling cost, we unify unlabeled sample selection and model training based on two principles. First, we exploit both labeled and unlabeled data using semi-supervised learning (SSL) to distill information from unlabeled data that improves representation learning and sample selection. Second, we propose a simple yet effective selection metric that is coherent with the training objective such that the selected samples are effective at improving model performance. Experimental results demonstrate superior performance of our proposed principles for limited labeled data compared to alternative AL and SSL combinations. In addition, we study an important problem – "When can we start AL?". We propose a measure that is empirically correlated with the AL target loss and can be used to assist in determining the proper start point.

## 1 Introduction

Deep learning models are improved when trained with more labeled data (Goodfellow et al., 2016). A standard deep learning development procedure involves constructing a large-scale labeled dataset and optimizing a model with it. Yet, in many real-world scenarios, large-scale labeled datasets can be very costly to acquire, especially when expert annotators are required, as in medical diagnosis and loan prediction. An ideal framework would integrate data labeling and model training to improve model performance with minimal amount of labeled data.

Active learning (AL) (Balcan et al., 2009) assists the learning procedure by judicious selection of unlabeled samples for labeling, with the goal of maximizing the ultimate model performance with minimal labeling cost. We focus on pool-based AL where an unlabeled data pool is given initially and the AL mechanism iteratively selects batches to label in conjunction with training. As the name "learning-based AL selection" suggests, each batch is selected with guidance from the previously-trained model, is labeled, and then added into the labeled dataset on which the model is trained.

Maximization of performance with minimal labeled data requires properly leveraging model learning and AL sample selection, especially in early AL cycles. While using the unlabeled pool as the candidates in the AL sample selection phase, it is natural for pool-based AL methods to integrate SSL objectives to improve performance by learning meaningful data representations from the unlabeled pool (Zhu et al., 2003; Tomanek & Hahn, 2009). However, fewer existing AL methods consider SSL during training (Drugman et al., 2019a; Rhee et al., 2017; Zhu et al., 2003; Sener & Savarese, 2018) compared to those utilizing only labeled samples. Moreover, we believe that AL selection criterion should be in coherence with the SSL objectives to select the most valuable samples, since (1) unsupervised losses could alter the learned representation and decision manifolds significantly (Oliver et al., 2018), and the AL sample selection should reflect that; (2) SSL already results in the embodiment of knowledge from unlabeled data in a meaningful way; thus AL selection should reflect the extra value of the labeled data on top of it. Motivated by these observations, we

propose an AL framework that combines SSL with AL and also a selection metric that is strongly related to the training objective.

In the absence of labeled data, a common practice to initiate AL is to uniformly select a small starting subset of data for labeling. Learning-based AL selection is then used in subsequent cycles. The size of the starting subset affects AL performance – when the start size is not sufficiently large, the models learned in subsequent AL cycles are highly-skewed and result in biased selection, a phenomenon commonly known as the *cold start problem* (Konyushkova et al., 2017; Houlsby et al., 2014). When cold start issues arise, learning-based selection yields samples that lead to lower performance improvement than using naive uniform sampling (Konyushkova et al., 2017). Increasing the start size alleviates the cold start problem, but consumes a larger portion of the labeling budget before learning-based AL selection is utilized. With better understanding of data, our method relieves this problem by allowing learning-based sample selection to be initialized from a much smaller start size. However, an ideal solution is determining a proper start size that is large enough in avoiding cold start problems, yet sufficiently small to minimize the labeling cost. To this end, we propose a measure that is empirically shown to be helpful in estimating the proper start size.

**Contributions:** We propose a simple yet effective selection metric that is in coherent with training objectives in SSL. The proposed AL method is based on an insight that has driven recent advances in SSL (Berthelot et al., 2019; Verma et al., 2019; Xie et al., 2019): a model should be consistent in its decisions between a sample and its meaningfully-distorted versions. This motivates us to introduce an AL selection principle: a sample along with its distorted variants that yields low consistency in predictions indicates that the SSL model is incapable of distilling useful information from that unlabeled sample, thus human labeling is needed. Experiments demonstrate that our proposed metric outperforms previous methods integrated with SSL. With various quantitative and qualitative analyses, we demonstrate the rationale behind why such a selection criteria is highly effective in AL. In addition, in an exploratory analysis we propose a measure that can be used to assist in determining the proper start size to mitigate cold start problems.

## 2 RELATED WORK

Extensive research has been done in AL (Dasgupta et al., 2008; Dasgupta & Hsu, 2008; Balcan et al., 2009; Cortes et al., 2019a). Traditional AL methods can be roughly classified into three categories: uncertainty-based methods, diversity-based methods and expected model change-based methods. Among uncertainty-based ones, methods based on *max entropy* (Lewis & Catlett, 1994; Lewis & Gale, 1994) and *max margin* (Roth & Small, 2006; Balcan et al., 2007; Joshi et al., 2009) are popular for their simplicity. Some other uncertainty-based methods measure distances between samples and the decision boundary (Tong & Koller, 2001; Brinker, 2003). Most uncertainty-based methods use heuristics, while recent work (Yoo & Kweon, 2019a) directly learns the target loss of inputs jointly with the training phase and shows promising results. Diversity-based methods select diverse samples that span the input space the most (Nguyen & Smeulders, 2004; Mac Aodha et al., 2014; Hasan & Roy-Chowdhury, 2015; Sener & Savarese, 2018). There are also methods that consider both uncertainty and diversity (Guo, 2010; Elhamifar et al., 2013; Yang et al., 2015). The third category estimates the future model status and selects samples that encourage optimal model improvement (Roy & McCallum, 2001; Settles et al., 2008; Freytag et al., 2014).

Both AL and SSL aim to improve learning with limited labeled data, thus they are naturally related. Only a few works have considered combining AL and SSL in different tasks. In Drugman et al. (2019b), joint application of SSL and AL is considered for speech understanding, and significant error reduction is demonstrated with limited labeled speech data. For AL, their selection criteria is based on a confidence score that quantifies the observed probabilities of words being correct. Rhee et al. (2017) propose an active semi-supervised learning system which demonstrates superior performance in the pedestrian detection task. Zhu et al. (2003) combine AL and SSL using Gaussian fields and validate their method on synthetic datasets. Sener & Savarese (2018) also consider SSL during AL cycles. However, in their setting, the performance improvement is marginal when adding SSL in comparison to their supervised counterpart, potentially due to the suboptimal SSL method.

Agreement-based methods, also referred as "query-by-committee", base the selection decisions on the opinions of a committee which consist of independent AL metrics or models (Seung et al., 1992; Cohn et al., 1994; McCallumzy & Nigamy, 1998; Iglesias et al., 2011; Beluch et al., 2018; Cortes et al., 2019b). Our method is related to agreement-based AL where samples are determined based

---

**Algorithm 1** A semi-supervised learning based AL framework

---

**Require:** Unlabeled data pool $\mathcal{D}$, the total number of steps $T$, AL batch size $K$, start size $K_0 \ll |\mathcal{D}|$
  $B_0 \leftarrow$ uniformly sampling from $\mathcal{D}$ with $|B_0| = K_0$
  $U_0 \leftarrow \mathcal{D} \backslash B_0$
  $L_0 \leftarrow \{(x, \mathcal{J}(x)) : x \in B_0\}$, where $\mathcal{J}(x)$ stands for the assigned label of $x$.
  **for** $t = 0, \ldots, T-1$ **do**
    (training) $M_t \leftarrow \arg\min_M \left\{ \frac{1}{|L_t|} \sum_{(x,y) \in L_t} \mathcal{L}_l(x, y, M) + \frac{1}{|U_t|} \sum_{x \in U_t} \mathcal{L}_u(x, M) \right\}$
    (selection) $B_{t+1} \leftarrow \arg\max_{B \subset U_t} \{\mathcal{C}(B, M_t), \ s.t. \ |B| = K\}$
    (labeling) $L_{t+1} \leftarrow L_t \cup \{(x, \mathcal{J}(x)) : x \in B_{t+1}\}$
    (pool update) $U_{t+1} \leftarrow U_t \backslash B_{t+1}$
  **end for**
  $M_T \leftarrow \arg\min_M \left\{ \frac{1}{|L_T|} \sum_{(x,y) \in L_T} \mathcal{L}_l(x, y, M) + \frac{1}{|U_T|} \sum_{x \in U_T} \mathcal{L}_u(x, M) \right\}$
  **return** $M_T$

---

on the conformity of different metrics or models. Specifically, our method selects data that mostly disagrees with the predictions of its augmentations.

## 3 CONSISTENCY-BASED SEMI-SUPERVISED AL

### 3.1 PROPOSED METHOD

We consider the setting of pool-based AL, where an unlabeled data pool is available for selection of samples to label. To minimize the labeling cost, we propose a method that unifies selection and model updates, overviewed in Algorithm 1. The proposed method has two key aspects.

Most conventional AL methods base model learning only on the available labeled data, which ignores the useful information in the unlabeled data. Our first contribution is incorporating a semi-supervised learning (SSL) objective in the training phases of AL. Specifically, as shown in Algorithm 1, each model $M_t$ is learned by minimizing an objective loss function of the form $\mathcal{L}_l + \mathcal{L}_u$. The model should both fit the labeled data well and obtain a good representations of the unlabeled data.

The design of the selection criterion plays a crucial role in integrating SSL and AL. To this end, our second contribution is a selection criterion $\mathcal{C}$ to better integrate AL selection mechanism in the SSL training framework.

It has been observed that predictions of deep neural networks are sensitive to small perturbations on the input data (Zheng et al., 2016; Azulay & Weiss, 2018). Recent successes in SSL (Athiwaratkun et al., 2019; Berthelot et al., 2019; Verma et al., 2019) are based on minimizing the notion of sensitivity to perturbations with the idea of inducing "consistency", i.e., imposing similarity in predictions when the input is perturbed in a way that would not change its perceptual content. For consistency-based semi-supervised training, a common choice of loss is $\mathcal{L}_u(x, M) = D(P(\hat{Y} = \ell | x, M), P(\hat{Y} = \ell | \tilde{x}, M))$, where $D$ is a distance function such as KL divergence (Xie et al., 2019), or L2 norm (Laine & Aila, 2017; Berthelot et al., 2019) and $\tilde{x}$ denotes a perturbation (augmentation) of the input $x$. Our proposal is motivated by the following intuition. First, the unsupervised objective exploits unlabeled data by encouraging consistent predictions across slightly distorted version of each unlabeled sample. Labeling samples with highly inconsistent predictions is valuable, since these samples are hard to be minimized using $\mathcal{L}_u$. Thus, they need human annotations to provide further useful supervision for model training. Second, the data that yields large model performance gain is not necessarily the data with the highest uncertainty, since neural network prefers learning with a particular curriculum (Bengio et al., 2009). The most uncertain data could be too hard to learn, and including them in training would be misleading. Thus, we argue that labeling samples that can be recognized to some extent but not consistently should benefit learning more compared to the most uncertain ones.

Specifically, we propose a simple metric $\mathcal{C}$ measures the inconsistency across perturbations. There are various ways to quantify consistency. Due to its empirically-observed superior performance, we

| Setting | Methods | # of labeled samples in total | | | |
|---|---|---|---|---|---|
| | | 100 | 150 | 200 | 250 |
| Supervised | Uniform | | 46.13±0.38 | 51.10±0.60 | 53.45±0.71 |
| | Entropy | 41.85 | 46.05±0.34 | 50.15±0.79 | 52.83±0.82 |
| | k-center | | 48.33±0.49 | 50.96±0.45 | 53.77±1.01 |
| Semi-supervised | Our method | 83.81 | **87.57±0.31** | **89.20±0.51** | **90.23±0.49** |

Table 1: Comparison with AL methods trained in supervised and semi-supervised setting on CIFAR-10. All methods start from 100 labeled samples (the third column). The following columns are results of different methods with the same selection batch size.

choose $\mathcal{C}(B, M) = \sum_{x \in B} \mathcal{E}(x, M)$, where

$$\mathcal{E}(x, M) = \sum_{\ell=1}^{J} \text{Var} \left[ P(\hat{Y} = \ell | x, M), P(\hat{Y} = \ell | \tilde{x}_1, M), ..., P(\hat{Y} = \ell | \tilde{x}_N, M) \right], \quad (1)$$

$J$ is the number of response classes and $N$ is the number of perturbed samples of the original input data $x$, $\{\tilde{x}_1, ..., \tilde{x}_N\}$, which can be obtained by standard augmentation operations [1]. Our method selects data samples with high $\mathcal{C}$ values for labeling, which may possess varying level of difficulty for the model to classify.

## 3.2 COMPARISONS WITH BASELINES

The practical performance of our method is demonstrated on two commonly used datasets: CIFAR-10 and CIFAR-100 (Krizhevsky et al., 2009) on the image classification task. Both datasets have 60K images in total, of which 10K images are for testing. CIFAR-10 consists of 10 classes and CIFAR-100 has 100 fine-grained classes. Different variants of SSL methods encourage consistency loss in different ways. In our implementation, we adopt the recently-proposed state-of-the-art method, Mixmatch (Berthelot et al., 2019), which proposes a specific loss term to encourage consistency. Following (Berthelot et al., 2019), we use Wide ResNet-28 (Oliver et al., 2018) with 32 filters as the base model and keep the default hyper-parameters for different settings from (Berthelot et al., 2019). In each cycle, $M_t$ is initialized with $M_{t-1}$. We select $K = 0.5 \cdot |L_0|$ samples for labeling by default. 50 augmentations of each image are obtained by horizontally flipping and random cropping, but we observe that 5 augmentations can produce comparable results. For a fair comparison, different selection methods start from the same initial model ($M_0$) and the reported results are over 5 trials.

We consider three representative selection methods for comparison:

- *Uniform* indicates random selection (no AL).
- *Entropy* is widely considered as an uncertainty-based baseline in previous methods (Sener & Savarese, 2018; Yoo & Kweon, 2019a). It selects uncertain samples that have maximum entropy of its predicted class probabilities.
- *k-center* (Sener & Savarese, 2018) selects representative samples by maximizing the distance between a selected sample and its nearest neighbor in the labeled pool. The feature from the last fully connected layer of the target model is used to calculate distances between samples.

As shown in Table 1, our method significantly outperforms the baseline methods which only learn from labeled data at each cycle. When 200 samples in total are labeled, our method outperforms *kcenter* by 39.24% accuracy. Next, we focus on comparing different methods in SSL framework. Figure 1 shows the effectiveness of our consistency-based selection in SSL setting by comparing with the baselines, when they are integrated into SSL. Our method outperforms baselines by a clear margin: on CIFAR-10, with 250 labeled images, our method outperforms *uniform* (passive selection) by $\sim 2.5\%$ and outperforms *k-center*, the state-of-the-art method, by $\sim 1.5\%$. As the number of labels increases, it is harder to improve model performance, but our method outperforms the *uniform* selection with 4K labels using only 2K labels, halving the labeled data requirements for the similar performance. Given access to all the labels (50K) for the entire training set, a fully-supervised model

---

[1] We follow https://github.com/google-research/mixmatch to perform data augmentation: the input images are randomly flipped and then randomly cropped.

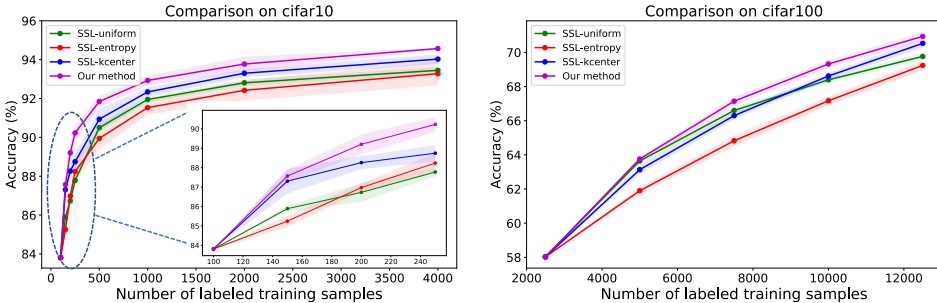

Figure 1: Model performance comparison with different sample selection methods on CIFAR-10 and CIFAR-100. Solid lines are average results over 5 trials. Shadows represent standard deviation.

| Methods | # of labeled samples in total | | | | |
| --- | --- | --- | --- | --- | --- |
| | 250 | 500 | 1000 | 2000 | 4000 |
| Uniform | 87.78±0.23 | 90.50±0.21 | 91.95±0.15 | 92.81±0.17 | 93.45±0.16 |
| Entropy | 88.24±0.51 | 89.95±0.58 | 91.53±0.35 | 92.42±0.53 | 93.28±0.61 |
| k-center | 88.75±0.42 | 90.94±0.53 | 92.34±0.24 | 93.30±0.21 | 94.03±0.25 |
| Our method | **90.23±0.39** | **91.84±0.29** | **92.93±0.26** | **93.78±0.38** | **94.57±0.06** |

Table 2: Comparison of different sampling methods in the SSL setting on CIFAR-10. Note that all the methods are under the SSL setting and start from 100 labeled samples. When the number of labeled samples reaches 250, AL batch size $K$ is set to be $|L_t|$.

achieves an accuracy of 95.83% (Berthelot et al., 2019). Our method with 4K examples has 30% more error compared to the fully supervised method. CIFAR-100 is a more challenging dataset as it has 10× more categories. On CIFAR-100, we observe a consistent outperformance of our method at all AL cycles.

### 3.3 ANALYSES OF CONSISTENCY-BASED SELECTION

To build insights on its superior performance, we analyze the samples selected by our method from several attributes, which are known to be important for AL.

**Uncertainty and overconfident mis-classification:** Uncertainty-based AL methods query the data samples close to the decision boundary. However, deep neural networks yield poorly-calibrated uncertainty estimates when the raw outputs are considered – they tend to be overconfident even when they are wrong (Guo et al., 2017; Lakshminarayanan et al., 2017). *entropy*-based AL metrics would not distinguished such overconfident mis-classifications, thus result in suboptimal selection. Figure 2 (left) demonstrates that our *consistency*-based selection is superior in detecting high-confident mis-classification cases than *entropy*. We use entropy to measure the uncertainty of the selected samples by different methods in Figure 2 (middle). It compares different approaches and shows that *uniform* and *k-center* methods do not base selection on uncertainty at all, whereas *consistency* tends to select highly-uncertain samples but not necessarily the top ones. Such samples should contribute to the performance gap with *entropy*. Figure 2 (right) illustrates some selected samples that are mis-classified with high confidence.

**Diversity:** Diversity has been proposed as a key factor for AL (Yang et al., 2015). *k-center* is a state-of-the-art AL method based on diversity (it prefers to select data points that span the whole input space). Towards this end, Figure 3 (right) visualizes the diversity of samples selected by different methods. We use principal components analysis to reduce the dimensionality of embedded samples to a two-dimensional space. *uniform* chooses samples equally-likely from the unlabeled pool. Samples selected by *entropy* are clustered in certain regions. On the other hand, *consistency* selects data samples as diverse as those selected by *k-center*. The average distances between top 1% samples selected by different methods are shown in Figure 3 (top-left). We can see that *entropy* chooses samples having small average distances, while *consistency* has a much larger average distance which is comparable to *uniform* and *k-center*.

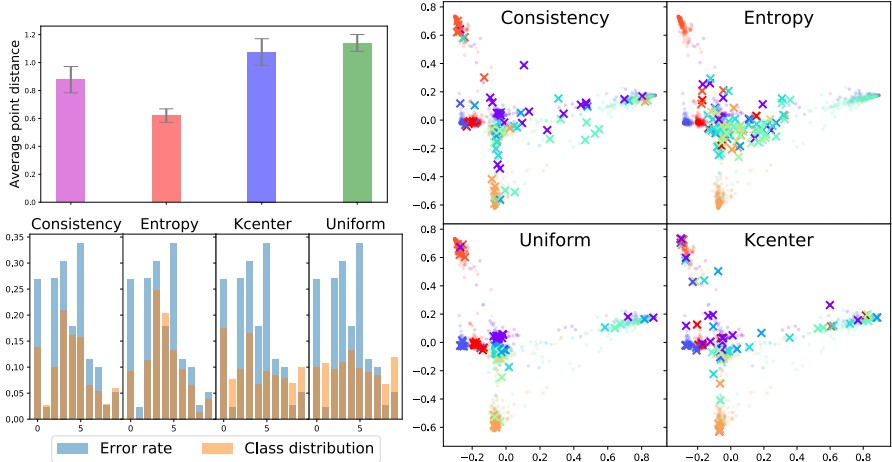

Figure 2: Left: Number of overconfident mis-classified samples in top $1\%$ samples ranked by different methods. Overconfident samples are defined as those having the highest class probability larger than threshold. Middle: the average entropy of unlabeled samples ranked by different selection metrics. The ranked samples are divided into 100 groups for computing average entropy. Right: Examples of overconfident mis-classified samples selected by *consistency*, which yield low entropy yet high inconsistency. Bird to horse indicates bird is mis-classified as horse.

Figure 3: Average distance between samples (top-left): the average pair-wise $L_2$ distance of top $1\%$ unlabeled samples ranked by different selection metrics. Per-class error rate vs. sample class distribution (bottom-left) are shown. Diversity visualization (right): Dots and crosses indicate unlabeled (un-selected) samples and the selected samples (top 100), respectively. Each color represent a ground truth class.

**Class distribution complies with classification error:** Figure 3 (bottom-left) shows the per-class classification error and the class distribution of samples selected by different metrics. Samples selected by *entropy* and *consistency* are correlated with per class classification error, unlike the samples selected by *uniform* and *k-center*.

## 4 WHEN CAN WE START LEARNING-BASED AL SELECTION?

### 4.1 COLD-START FAILURE

When the size of the initial labeled dataset is too small, the learned decision boundaries could be skewed and AL selection based on the model outputs could be biased. To illustrate the problem, Figure 4 shows the toy two-moons dataset using a simple support vector machine (in supervised setting with the RBF kernel) to learn the decision boundary (Oliver et al., 2018). As can be seen, the naive uniform sampling approach achieves better predictive accuracy by exploring the whole space. On the other hand, the samples selected by *max entropy* concentrate around a poorly learned boundary. In another example, we study the effects of cold start using deep neural networks on CIFAR-10, shown in Figure 5. Using uniform sampling to select different starting sizes, AL methods achieve different predictive accuracy. For example, the model starting with $K_0 = 50$ data points clearly under-performs the model starting with $K_0 = 100$ samples, when both models reach 150

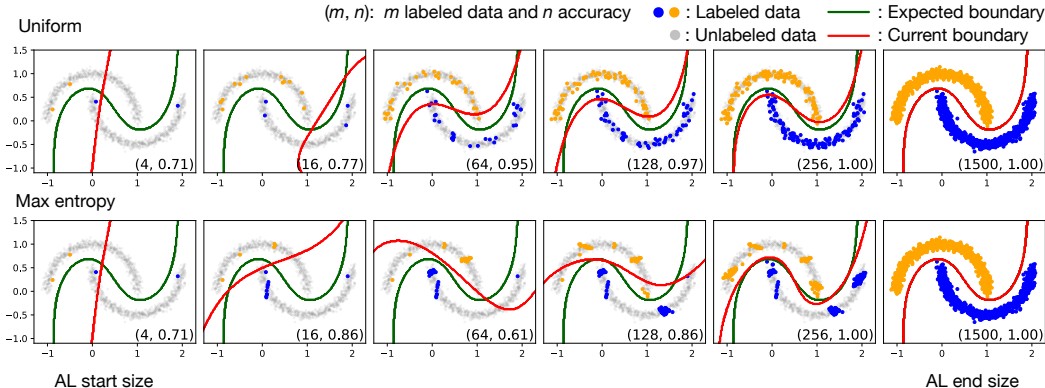

Figure 4: Illustration of cold-start problems for uncertainty-based AL. The start size and end size has the same labeled data. When the learned decision boundary is far away from the expected boundary (the boundary when all labels are available for the entire training set), e.g. the second and third columns, the selected samples by uncertainty-based AL is biased.

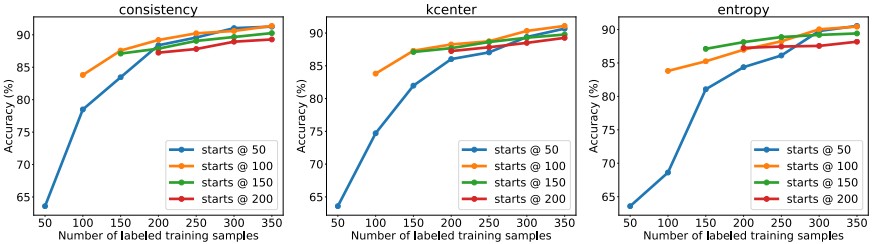

Figure 5: Performance comparison of different sampling methods trained with SSL framework on CIFAR-10 when AL starts from different number of labeled samples.

labeled samples. It may due to the cold start problem encountered when $K_0 = 50$. While, given a limited labeling budget, naively choosing a large start size is also not practically desirable, because it may lead to under-utilization of learning-based selection. For example, our method starting from $K_0 = 100$ labeled samples has better performance than starting from 150 or 200, since we have more AL cycles in the former case given the same label budget.

The semi-supervised nature of our learning proposal encourages the practice of initiating learning-based sample selection from a much smaller start size. However, the learned model can still be skewed at extreme early AL stages. These observations motivate us to propose a systematic way of inferring a proper starting size. We analyze this problem and propose an approach to assist in determining the start size in practice.

## 4.2 AN EXPLORATORY ANALYSIS IN START SIZE SELECTION

Recall from the last step of Algorithm 1, if $T$ is set such that $U_T = \emptyset$, i.e., if the entire dataset has been labeled, then the final model $M_T$ is trained to minimize the purely supervised loss $\mathcal{L}_l$ on the total labeled dataset $L_T$. Consider the cross-entropy loss function for any classifier $p(\hat{Y}|X)$, which we call the *AL target loss*:

$$\mathcal{L}_l \left[ L_T, p(\hat{Y}|X) \right] = -\frac{1}{|L_T|} \sum_{(x,y) \in L_T} \log p(\hat{Y} = y | X = x). \quad (2)$$

Note that the goal of an AL method can be viewed as minimizing the AL target loss with a small subset of the entire training set $L_T$ (Zhu et al., 2003). In any intermediate AL step, we expect the loss on the current labeled subset to mimic the target loss. If cold start problems occur, the model does a poor job in approximating and minimizing equation 2. The quality of the samples selected

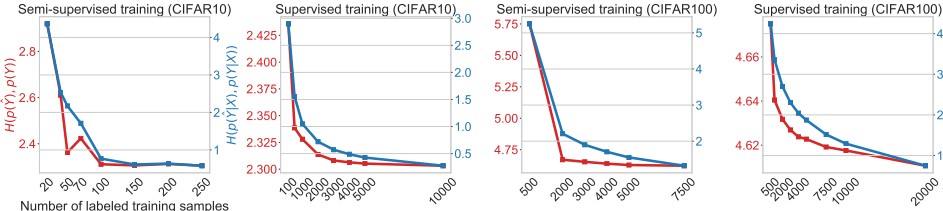

Figure 6: Empirical risk (i.e. the target loss) on the entire training data (in blue) and cross-entropy between $p(\hat{Y})$ and $p(Y)$ show strong correlations in both semi-supervised and supervised settings.

in the subsequent AL cycles would be consequently poor. Therefore, it is crucial to understand the performance of the currently-learned model in minimizing the criterion in equation 2. However, since the labeled data set $L_t$ at cycle $t$ is a strict subset of the total training set $L_T$, it is impossible to simply plug the most recently learned model $\hat{Y}$ in equation 2 for direct calculation.

Our approximation to the target loss is based on the following proposition, which gives upper and lower bounds on the expected loss, to which the target loss approximates:

**Proposition 1.** *For any given distribution of $Y$, and any learned model $\hat{Y}$, we have*

$$H\left[p(Y), p(\hat{Y})\right] - H[p(X)] \leq R_H\left[p(\hat{Y}|X)\right] = \mathbb{E}_X\left\{H\left[p(Y|X), p(\hat{Y}|X)\right]\right\}$$
$$\leq H\left[p(Y), p(\hat{Y})\right] - H[p(X)] - \log \hat{Z}, \quad (3)$$

*where $H[p, q]$ is the cross-entropy between two distributions $p$ and $q$, $H[p(X)]$ is the entropy of the random variable $X$, and $\hat{Z} = \min_{x,y} p(X = x|\hat{Y} = y)$.*

Proposition 1 indicates that the expected cross-entropy loss can be both upper and lower bounded. In particular, both bounds involve the quantity $H[p(Y), p(\hat{Y})]$, which suggests that $H[p(Y), p(\hat{Y})]$ could potentially be tracked to analyze $R_H[p(\hat{Y}|X)]$ for different numbers of samples. Unlike the unavailable target loss on the entire training set, $H[p(Y), p(\hat{Y})]$ does not need all data to be labeled. In fact, to compute $H[p(Y), p(\hat{Y})]$, we just need to specify a distribution for $Y$, which could be assumed from prior knowledge or estimated using all of the labels in the starting cycle.

In Figure 6, we observe a strong correlation between the target loss and $H[p(Y), p(\hat{Y})]$, where $Y$ is assumed to be uniform. We see how $H[p(Y), p(\hat{Y})]$ can be used to identify the trend when the actual target is minimized. Particularly, in SSL setting, a practitioner may set the starting set size to 100 or 150 labeled samples on CIFAR-10, as the value of $H[p(Y), p(\hat{Y})]$ essentially ceases decreasing, which coincide with the oracle stopping points if we were given access to the target loss. In contrast, a start size of 50 has much higher $H[p(Y), p(\hat{Y})]$, which leads to less favorable performance. A similar pattern in the supervised learning setting is shown in Figure 6.

## 5 CONCLUSION AND FUTURE WORK

We presented a simple pool-based AL selection metric to select data for labeling by leveraging unsupervised information of unlabeled data during training. Experiments show that our method outperforms previous state-of-the-art AL methods under the SSL setting. Our proposed metric implicitly balances uncertainty and diversity when making selection. The design of our method focuses on the principles of consistency in SSL. For alternative SSL methods based on other principles, it is necessary to revisit AL selection with respect to their training objectives, which will be considered in future work. In addition, we study and address a very practically valuable yet challenging question — "When can we start learning-based AL selection?". We present a measure to assist in determining proper start size. Experimental analysis demonstrates that the proposed measure correlates well with the AL target loss (i.e. the ultimate the supervised loss on all labeled data). In practice, it can be tracked to examine the model without requesting a large validation set.

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

APPENDIX

## A    PROOF OF PROPOSITION 1

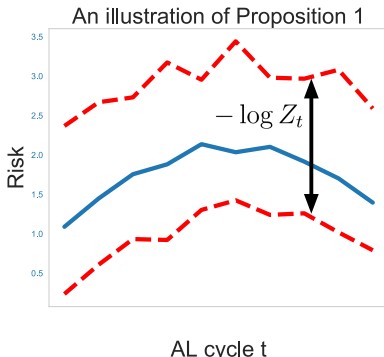

Figure A1: An illustration of Proposition 1: the blue curve represents the (expected) cross-entropy, and the two red curves are the lower and upper bounds. The value $-\log \hat{Z}_t$ characterizes the range of the bounds.

*Proof.* Denote $\mathcal{X}$ as the feature space and $\{1, \dots, J\}$ as the label space. Note that by Baye's formula and the law of total probability, we have

$$R_H[p(\hat{Y}|X)] = \mathrm{E}_X \left\{ H \left[ p(Y|X), p(\hat{Y}|X) \right] \right\}$$

$$= -\sum_{x \in \mathcal{X}} \sum_{y=1}^{J} p(Y = y|X = x) \log p(\hat{Y} = y|X = x) p(X = x)$$

$$= -\sum_{y=1}^{J} \sum_{x \in \mathcal{X}} p(X = x, Y = y) \log \left[ \frac{p(\hat{Y} = y) p(X = x|\hat{Y} = y)}{p(X = x)} \right]$$

$$= -\sum_{y=1}^{J} \sum_{x \in \mathcal{X}} p(X = x, Y = y) \log p(\hat{Y} = y) - \sum_{y=1}^{J} \sum_{x \in \mathcal{X}} p(X = x, Y = y) \log \left[ \frac{p(X = x|\hat{Y} = y)}{p(X = x)} \right]$$

$$= -\sum_{y=1}^{J} p(Y = y) \log p(\hat{Y} = y) - \sum_{x \in \mathcal{X}} \sum_{y=1}^{J} p(X = x, Y = y) \log \left[ p(X = x|\hat{Y} = y) \right]$$

$$+ \sum_{x \in \mathcal{X}} \sum_{y=1}^{J} p(X = x, Y = y) \log \left[ p(X = x) \right]$$

$$= H \left[ p(Y), p(\hat{Y}) \right] + \sum_{x \in \mathcal{X}} p(X = x) \log \left[ p(X = x) \right] - \sum_{x \in \mathcal{X}} \sum_{y=1}^{J} p(X = x, Y = y) \log \left[ p(X = x|\hat{Y} = y) \right]$$

$$= H \left[ p(Y), p(\hat{Y}) \right] - H \left[ p(X) \right] - \sum_{x \in \mathcal{X}} \sum_{y=1}^{J} p(X = x, Y = y) \log \left[ p(X = x|\hat{Y} = y) \right]. \tag{4}$$

We first give a lower bound. Note that $p(X = x|\hat{Y} = y) \leq 1$ for any $(x, y) \in \mathcal{X} \times [J]$, so equation 4 implies that

$$\mathrm{E}_X \left\{ H \left[ p(Y|X), p(\hat{Y}|X) \right] \right\} \geq H \left[ p(Y), p(\hat{Y}) \right] - H \left[ p(X) \right].$$

| Methods | # of labeled samples in total | | | | |
|---------|------|-------|------|------|------|
| | 1000 | 1500 | 2000 | 2500 | 3000 |
| Uniform | | 75.38±0.17 | 77.46±0.3 | 78.79±0.38 | 80.81±0.28 |
| Entropy | 72.93 | 76.31±0.18 | 79.50±0.29 | 81.30±0.31 | 82.67±0.55 |
| k-center | | 74.25±0.29 | 77.56±0.30 | 79.50±0.20 | 81.70±0.32 |
| Our method | | 76.63±0.17 | 79.39±0.31 | 80.99±0.39 | 82.75±0.26 |

Table 3: Comparison of different sampling methods in the supervised setting on CIFAR-10. All methods start from 1000 labeled samples.

To prove the upper bound, denote $\min_{(x,y)\in\mathcal{X}\times[J]} p(X=x|\hat{Y}=y) = \hat{Z} \in (0,1)$ where $(x,y) \in \mathcal{X} \times [J]$. Then from equation 4

$$\mathrm{E}_X\left\{H\left[p(Y|X), p(\hat{Y}|X)\right]\right\} \leq H\left[p(Y), p(\hat{Y})\right] - H\left[p(X)\right] - \log\hat{Z}\sum_{x\in\mathcal{X}}\sum_{y=1}^{J} p(X=x, Y=y)$$

$$= H\left[p(Y), p(\hat{Y})\right] - H\left[p(X)\right] - \log\hat{Z}.$$

$\square$

## B   MORE DISCUSSION

### B.1   CONSISTENCY-BASED AL IN SUPERVISED LEARNING

We also curious about how well our method performs under supervised learning using only labeled samples. Following Yoo & Kweon (2019b), we start with 1000 labeled samples on CIFAR-10. As shown in Table 3, after 4 AL cycles ($B = 500$, totaling 3000 labels), *uniform*, *k-center*, *entropy* and our method (*consistency*) achieve accuracy of 80.81%, 81.70%, 82.67% and 82.75%, respectively. It shows that *consistency* still works even if our model is trained using only labeled samples. However, the improvement of *consistency* compared to other baseline methods (especially *entropy*) is marginal.

### B.2   OUT-OF-DISTRIBUTION AND CHALLENGING SAMPLES

In real-world scenarios, it is very likely that not all unlabeled data are irrelevant to the task. Therefore, if a sample remains high uncertainty given arbitrary perturbations, it is probably a out-of-distribution example (Lee et al., 2018). In addition, selecting the hardest samples are not preferred, because it could be "over-challenging" for current model as suggested by the study of curriculum learning (Bengio et al., 2009). It can be easily inferred that our proposed selection can avoid such cases (see equation 1). More exploration of active learning with out-of-distribution samples is left for future work.

