# OpenReview forum: "Consistency-Based Semi-Supervised Active Learning: Towards Minimizing Labeling Budget"
_ICLR.cc/2020/Conference — Reject_

### Official Review · AnonReviewer2 · 2019-10-21
**Official Blind Review #2**

**Rating:** 6

**Review:**

This paper proposes a semi-supervised active learning method to reduce the labeling cost. In the proposed method, a selection criterion to better integrate AL selection mechanism in SSL training framework is designed. The simple metric that aims to measure the inconsistency across a certain number of meaningful perturbations. It considers N perturbed samples of the original input data x, which can be obtained by standard augmentation operations (e.g. random crops and horizontal flips for image data). Then the variance is adopted to quantify consistency.  In this way, the proposed method prefers data samples with high values, which may possess varying level of difficulty for the model to classify. To verify the effectiveness of the proposed method, several baseline methods are compared on several benchmark data sets, and the proposed method has achieved better performance. Meanwhile, to deal with the “cold start” problem, a measure that is found to be empirically correlated with the AL target loss is proposed, and this measure can be used to assist in determining the proper start size. However, there are some minor concerns:
[1] The consistency of a sample is measured based on the perturbed samples. How to generate these perturbed samples may have a great influence on the query results. In the paper, it said that these samples are generated by standard augmentation operations (e.g. random crops and horizontal flips for image data). This representation is hard to follow in the experiments. If possible, it is better to show in details.
[2] In the uncertainty of active learning, the samples are selected from different distributions in the unlabeled data, for example, the marginal sampling selects the samples around the classification hyperlanes (Settles, Burr. "Active learning." Synthesis Lectures on Artificial Intelligence and Machine Learning 6.1 (2012): 1-114.). Can you show which samples may be selected in the unlabeled data. In this way, the proposed criterion can be followed more easily.
[3] In the experiments, whether the proposed method can select a batch of samples at each iteration. How about the influence of the batch size.


**Experience Assessment:**

I have published in this field for several years.

**Review Assessment: Checking Correctness Of Derivations And Theory:**

I assessed the sensibility of the derivations and theory.

**Review Assessment: Checking Correctness Of Experiments:**

I did not assess the experiments.

**Review Assessment: Thoroughness In Paper Reading:**

I read the paper at least twice and used my best judgement in assessing the paper.

---

> ### Author Response · Authors · 2019-11-10
> **(Updated) Thanks for your thoughtful suggestions.**
>
> Thanks for your thoughtful comments. Please see our responses as follows.
>
> Q1. The consistency of a sample is measured based on the perturbed samples. How to generate these perturbed samples may have a great influence on the query results. In the paper, it said that these samples are generated by standard augmentation operations (e.g. random crops and horizontal flips for image data). This representation is hard to follow in the experiments. If possible, it is better to show in details.
>
> Our response: We augment the input images following (Berthelot et al., 2019). The input images are randomly flipped (left-right-random-flip code link: https://github.com/google-research/mixmatch/blob/1011a1d51eaa9ca6f5dba02096a848d1fe3fc38e/libml/data.py#L87) and then randomly cropped  (code link: https://github.com/google-research/mixmatch/blob/1011a1d51eaa9ca6f5dba02096a848d1fe3fc38e/libml/data.py#L91).
>
> We have added these details in the updated version.
>
> Q2. In the uncertainty of active learning, the samples are selected from different distributions in the unlabeled data, for example, the marginal sampling selects the samples around the classification hyperlanes (Settles, Burr. "Active learning." Synthesis Lectures on Artificial Intelligence and Machine Learning 6.1 (2012): 1-114.). Can you show which samples may be selected in the unlabeled data. In this way, the proposed criterion can be followed more easily.
>
> Our response: It is a great idea to analyze the selected samples. We analyze the selected samples and compare with those selected by other methods in Figure 2 and Figure 3.
>
> Figure 2 shows our selected samples tend to have high average entropy (uncertainty). On the other hand, uncertain samples are not always informative. For example, they may also promote focusing outlier samples outside the distribution of data which may harm the model performance in some cases. As shown in Figure 2 (middle), top-ranked samples chosen by our AL method can sometimes have low entropy as well - our metric considers uncertainty, but can also deviate from it in some cases. Figure 3 underlines that our method tends to focus on diversity in selection. These results suggest that the selected samples consider a combination of uncertainty and diversity attributes in different regimes.
>
> Q3. In the experiments, the proposed method can select a batch of samples at each iteration. How about the influence of the batch size.
>
> Our response: There is always a trade-off between a large AL batch size and a small AL batch size. Ideally, we would like to use AL as much as possible. Selecting a very large batch of samples will lead to insufficient usage of active learning given a limited budget. However, a very small AL batch size would lead to much more AL cycles which is computationally expensive.
>
> Our method is effective using reasonable AL batch sizes. We conducted more experiments on CIFAR-10 under the setting of Figure 1 (starting from 100 labeled samples), using different AL batch sizes. Our experiments show that when labeling 200 samples in total, we obtain accuracy of 89.5%, 89.2% and 89.3% when AL batch size is set to be 25, 50 and 100, respectively. These results suggest that, the performances are comparable using reasonable AL batch sizes.
>
> ----------------------------------------------------------------------------------------------------------------
> |                     AL batch size                                |        25       |         50        |       100       |
> ----------------------------------------------------------------------------------------------------------------
> |# of AL cycles to reach 200 labeled data      |         4        |          2         |          1         |
> ----------------------------------------------------------------------------------------------------------------
> |                             Accuracy                                |      89.5%  |       89.2%    |        89.3%  |
> ----------------------------------------------------------------------------------------------------------------

---

### Official Review · AnonReviewer1 · 2019-10-22
**Official Blind Review #1**

**Rating:** 6

**Review:**

This paper proposes a new combination method for active learning and semi-supervised learning, where the objective is to make predictions that are robust to perturbations (for SSL) and select points for labeling with labels that differ under perturbations. This technique achieves 2x label efficiency over SSL with uniform-random sampling. Additionally, the authors assess (at least for CIFAR-10 with batch size 50) the best starting random seed set as 100 labels, known as K_0 in this work. This work yields pretty good empirical results and has a conceptually unified approach to SSL and active learning building off of recent works.


Comments:

 - This paper compares in Table 1 the difference between just active learning vs active learning + SSL. I'm not sure this is a fair comparison. I think the better comparison is shown in Table 2.

 - The authors write that "when only 100 samples are labeled, our method outperforms kcenter by 39.24% accuracy". Do the authors mean after 100 additional labels are acquired (so 200 labels) or is this number off?

 - Can the authors clarify what is meant by "or some labels correspond to rare cases, as in self-driving cars"? Why are such datasets more costly to acquire? Is it because of the size of the self-driving car datasets?

 - Although the method is more unified than some other AL + SSL approaches, I wonder if the L_u(x,M) can be made to look more like the C(B,M) = \sum E(x,M). In particular, L_u(x,M) uses just a single perturbation and a different "distance" function than E(x,M) which uses N perturbations.

 - The authors state that they lose 1.26% accuracy to the fully supervised model. However, this is very much not within the margin of measurement error and 1.26% accuracy is rather significant for accuracies around 95%. Another way of putting it is that the method in the paper with 4K examples has 30% more error compared to the fully supervised method. Can the authors either change this claim or provide a number of labels where their method achieves the fully-supervised accuracy?




**Experience Assessment:**

I have published one or two papers in this area.

**Review Assessment: Checking Correctness Of Derivations And Theory:**

I assessed the sensibility of the derivations and theory.

**Review Assessment: Checking Correctness Of Experiments:**

I assessed the sensibility of the experiments.

**Review Assessment: Thoroughness In Paper Reading:**

I read the paper at least twice and used my best judgement in assessing the paper.

---

> ### Author Response · Authors · 2019-11-10
> **(Updated) Thanks for your valuable suggestions.**
>
> Thanks for your valuable  comments. Please see our responses as follows.
>
> Q1. This paper compares in Table 1 the difference between just active learning vs active learning + SSL. I'm not sure this is a fair comparison. I think the better comparison is shown in Table 2.
>
> Our response: We agree with you that the Table 2 presents the fair comparison among different selection metrics. As we mentioned in the introduction, many existing active learning methods train their models using only labeled samples at each AL cycle. The purpose of Table 1 is to motivate our work by showing the effectiveness of involving SSL in active learning, in other words demonstrating the value of using unlabeled data for learning.
>
> Q2. The authors write that "when only 100 samples are labeled, our method outperforms kcenter by 39.24% accuracy". Do the authors mean after 100 additional labels are acquired (so 200 labels) or is this number off?
>
> Our response: Thanks for pointing this out. It is after 100 additional labels are acquired. We have clarified this in the updated version.
>
> Q3. Can the authors clarify what is meant by "or some labels correspond to rare cases, as in self-driving cars"? Why are such datasets more costly to acquire? Is it because of the size of the self-driving car datasets?
>
> Our response: Self-driving car datasets have long-tail distribution, generally consisting of vast amount of normal driving conditions, and various rare cases, e.g., children playing on the road and pedestrian lying on the street. When data samples are uniformly sampled to be labeled, to obtain enough samples to represent rare cases, one would need to label abundantly high number of examples for common cases. AL method is potentially useful in this scenario where it can make labeling suggestions prone to rare cases.
>
> The original statement was confusing and did not convey much information, so we decided to remove the sentence in the current version.
>
> Q4. Although the method is more unified than some other AL + SSL approaches, I wonder if the L_u(x,M) can be made to look more like the C(B,M) = \sum E(x,M). In particular, L_u(x,M) uses just a single perturbation and a different "distance" function than E(x,M) which uses N perturbations.
>
> Our response: We agree with the reviewer that C(B,M) = \sum E(x,M) can be an interesting replacement for L_u(x,M) during model training to encourage unsupervised consistency. However, estimating accurate variance usually requires a large amount of samples (N=50 in our case). It will significantly increase the computation cost for model training (since every perturbation needs to forward the model once). We would like to discover this option in the future work.
>
> Q5. The authors state that they lose 1.26% accuracy to the fully supervised model. However, this is very much not within the margin of measurement error and 1.26% accuracy is rather significant for accuracies around 95%. Another way of putting it is that the method in the paper with 4K examples has 30% more error compared to the fully supervised method. Can the authors either change this claim or provide a number of labels where their method achieves the fully-supervised accuracy?
>
> Our response: Thanks for your suggestion! We have changed this claim as suggested to “our method with 4K examples has 30% more error compared to the fully supervised method”.

---

### Public Comment · ~Hao_Zhongkai1 · 2019-10-30
**Confused about the training strategy**

I'm confused that when you select a new batch from the unlabeled data pool, you need to minimize the loss on the labeled dataset. Do you only finetune your neural network on the new batch or finetune on the whole labeled dataset, or completely retrain a model from random initialization? I notice that many works on active learning did not mention the training or finetuning strategy. When the dataset is large, completely retraining a deep neural network is time-consuming.

---

> ### Author Response · Authors · 2019-10-30
> **Reply to "Confused about the training strategy"**
>
> Thanks for your interest.
>
> We finetune the model on the whole dataset. Our model is trained in the semi-supervised mode, so in each cycle both labeled and unlabeled data is used for training.

---

> > ### Public Comment · ~Hao_Zhongkai1 · 2019-10-31
> > **Does finetuning epochs affect the accuracy?**
> >
> > Many thanks for your response. I noticed that when the dataset is small, which is exactly the situation when active learning begins, training and finetuning on the small dataset might be easy to overfit. I'm confused that does the selection of the finetuning epochs number affect the accuracy on the test dataset?

---

> > > ### Author Response · Authors · 2019-11-03
> > > **Reply to "Does finetuning epochs affect the accuracy?"**
> > >
> > > Standard supervised training is easily overfitted on small labeled training data. Finetuning on an overfitted model can hurt the performance based on our empirical knowledge. Based on our observation, the advanced semi-supervised method is stable and robust to overfitting.  That is also our motivation, which is taking advantage of semi-supervised training to significantly improve active learning at early AL cycles.

---

### Public Comment · ~Christoph_Mayer1 · 2019-11-07
**Some thoughts and questions**

I got some questions about the novelty of the paper and about related work.

I am wondering whether the idea of combining SSL and AL was not already introduced by others (Sener et al, CEAL from Wang et al.)? Nonetheless your performance is much better I guess mainly because of Mixmatch? How about other SSL techniques (mean teacher, VAT, SNTG or GAN based methods) does your approach also work there?

Does you k-center AL (Sener et al.) include the MIP (robust k-center) or is it just plain k-center?

The consistency based AL criterion is indeed interesting but I think besides validating on SSL it should also be tested on standard supervised learning. I noticed such experiments in the supplementary is performs as good as entropy sampling. How does it compare to more recent approaches such as Learning Loss for Active Learning (CVPR19) or Bayesian Generative Active Deep Learning (ICML19). I think these experiments are very important for the proposed AL criterion to know in which setting it should be used.  Why are they not part of the main paper?

Another thing that I am wondering about is did you do experiments for Learning Loss for Active Learning or Bayesian Generative Active Deep Learning using a SSL approach to train the classifier? How do they perform in SSL scenario?

I think the section 4 is interesting. However, I believe that it would be optimal to do active learning as early as possible optimally we should never use random sampling but select already samples at the beginning. The plots in Fig. 3. show that it the performance it not very good if we start too early with AL with too few labeled samples for the three strategies. So it means that if we are forced to start with 50 labeled samples it would be a good idea to select the next 50 samples randomly right? So I get the feeling that the studied three criteria are just bad in this situation but in general we should start as early as possible but we need other criteria. What are your thoughts about this? Maybe it would also be a good idea to vary the number of labeled samples that we add in each AL cycle.

Thanks for your answer.

---

> ### Author Response · Authors · 2019-11-10
> **Thanks for your interest.**
>
> Thanks for your interest in several parts of our work and your kind suggestions to the possible future extensions of this work. Please see our responses as follows.
>
> 1. I am wondering whether the idea of combining SSL and AL was not already introduced by others (Sener et al, CEAL from Wang et al.)? Nonetheless your performance is much better I guess mainly because of Mixmatch? How about other SSL techniques (mean teacher, VAT, SNTG or GAN based methods) does your approach also work there?
>
> Our response: As we mentioned in the related work, there are existing AL work incorporating SSL. Our contribution (novelty) is unifying the unlabeled sample selection and the SSL model training. Also, we propose consistency based AL criterion, which is simple and effective. In Tab. 2, we compared with other metrics to demonstrate the effectiveness of our proposed metric when all methods are based on Mixmatch.
>
> As we mentioned in the section 5, regarding to other SSL methods that are not based on consistency principles, it might be necessary to redesign AL selection criteria aligned with their training objectives. We leave this investigation to future research.
>
> 2. Does you k-center AL (Sener et al.) include the MIP (robust k-center) or is it just plain k-center?
>
> Our response: We used plain k-center.
>
> 3. The consistency based AL criterion is indeed interesting but I think besides validating on SSL it should also be tested on standard supervised learning. I noticed such experiments in the supplementary is performs as good as entropy sampling. How does it compare to more recent approaches such as Learning Loss for Active Learning (CVPR19) or Bayesian Generative Active Deep Learning (ICML19). I think these experiments are very important for the proposed AL criterion to know in which setting it should be used.  Why are they not part of the main paper?
>
> Our response: Thanks for your interest in our consistency based AL criterion.
> It is not part of the main paper, because using SSL in AL is much more effective compared to using supervised training (shown in Tab 1) and our contribution is proposing a framework which unifies unlabeled data selection and SSL. Testing our selection metric on different supervised models is not the focus of this work.
>
> 4. Another thing that I am wondering about is did you do experiments for Learning Loss for Active Learning or Bayesian Generative Active Deep Learning using a SSL approach to train the classifier? How do they perform in SSL scenario?
>
> Our response: Converting these supervised models to SSL approaches might be an interesting future work, but it is beyond the scope of this work. Our focus is unifying unlabeled data selection and SSL model training.
>
> 5. I think the section 4 is interesting. However, I believe that it would be optimal to do active learning as early as possible optimally we should never use random sampling but select already samples at the beginning. The plots in Fig. 3. show that it the performance it not very good if we start too early with AL with too few labeled samples for the three strategies. So it means that if we are forced to start with 50 labeled samples it would be a good idea to select the next 50 samples randomly right? So I get the feeling that the studied three criteria are just bad in this situation but in general we should start as early as possible but we need other criteria. What are your thoughts about this? Maybe it would also be a good idea to vary the number of labeled samples that we add in each AL cycle.
>
> Our response: This is an interesting future work. A potential idea can be switching from AL methods to random whenever it is needed. Intuitively, random selection will be chosen more at the beginning and less afterwards. Another potential idea is AL with varying AL batch sizes (e.g. start with large batch size and then decrease for steep performance increase and then decrease again etc.).

---

### Author Response · Authors · 2019-11-15
**Updates of the manuscript**

Thanks again for all the valuable comments. We have updated our manuscript.
The current version includes the following changes.

1. We improved our writing (fixed typos and grammar issues etc.).
2. We revised some confusing statements.
3. We revised the manuscript according to Q2, Q3 and Q5 of the Reviewer 1.
4. We revised the manuscript according to Q1 of the Reviewer 2.

---

### Decision · Program_Chairs · 2019-12-19

**Decision:**

Reject

**Comment:**

The authors leverage advances in semi-supervised learning and data augmentation to propose a method for active learning. The AL method is based on the principle that a model should consistently label across perturbation/augmentations of examples, and thus propose to choose samples for active learning based on how much the estimated label distribution changes based on different perturbations of a given example. The method is intuitive and the experiments provide some evidence of efficacy. However, during discussion there was a lingering question of novelty that eventually swayed the group to reject this paper.